EnsemV3X: a novel ensembled deep learning architecture for multi-label scene classification

Sobti Priyal 1
http://orcid.org/0000-0002-9821-6146 Nayyar Anand 2 anandnayyar@duytan.edu.vn
Niharika 1
Nagrath Preeti 1
1 Department of Computer Science and Engineering, Bharati Vidyapeeth's College of Engineering , New Delhi , India
2 Graduate School/Faculty of Information Technology, Duy Tan University , Da Nang , Viet Nam
Gao Zhiwei
Electronic publication date: 2021 May 25
Publication date: 2021
Volume: 7
Electronic Location ID: e557
Received 2020 Oct 21; Accepted 2021 May 1
Copyright: © 2021 Sobti et al.
Copyright year: 2021
Copyright holder: Sobti et al.
License: This is an open access article distributed under the terms of the Creative Commons Attribution License, which permits unrestricted use, distribution, reproduction and adaptation in any medium and for any purpose provided that it is properly attributed. For attribution, the original author(s), title, publication source (PeerJ Computer Science) and either DOI or URL of the article must be cited.
License URL: https://creativecommons.org/licenses/by/4.0/

Keywords: EnsemV3X, Multi-label scene classification, Convolutional Neural Network (CNN), Image classification, Deep learning

Funding: The authors received no funding for this work.

==============================
Convolutional neural network is widely used to perform the task of image classification, including pretraining, followed by fine-tuning whereby features are adapted to perform the target task, on ImageNet. ImageNet is a large database consisting of 15 million images belonging to 22,000 categories. Images collected from the Web are labeled using Amazon Mechanical Turk crowd-sourcing tool by human labelers. ImageNet is useful for transfer learning because of the sheer volume of its dataset and the number of object classes available. Transfer learning using pretrained models is useful because it helps to build computer vision models in an accurate and inexpensive manner. Models that have been pretrained on substantial datasets are used and repurposed for our requirements. Scene recognition is a widely used application of computer vision in many communities and industries, such as tourism. This study aims to show multilabel scene classification using five architectures, namely, VGG16, VGG19, ResNet50, InceptionV3, and Xception using ImageNet weights available in the Keras library. The performance of different architectures is comprehensively compared in the study. Finally, EnsemV3X is presented in this study. The proposed model with reduced number of parameters is superior to state-of-of-the-art models Inception and Xception because it demonstrates an accuracy of 91%.

Introduction

Many problems, such as transferring our findings to a new dataset (Goyal & Benjamin, 2014), object detection (Ren et al., 2015), scene recognition (Herranz, Jiang & Li, 2016), have been extensively investigated due to network architectures measured against the ImageNet dataset (Deng et al., 2009; Krizhevsky, Sutskever & Hinton, 2012). Furthermore, any architecture that performs effectively on ImageNet is assumed effectual on other computer vision tasks as well (Voulodimos et al., 2018). If the second to last layers demonstrate satisfactory features, then networks, such as ImageNet, show acceptable performance with transfer learning (Kornblith, Shlens & Le, 2019). ImageNet is an extremely diverse dataset comprising of more than 15 million images that belong to 22,000 categories (Krizhevsky, Sutskever & Hinton, 2012) which are structured according to WordNet hierarchy (Miller, 1995). WordNet contains more than 100,000 multiple words or phrases called “synsets” or “synonym sets,” and ImageNet attempts to provide 1,000 images for each synset (Miller, 1998). ImageNet aims to provide researchers with sophisticated resources for computer vision. An annual competition named ImageNet Large-Scale Visual Recognition Challenge (ILSVRC) (Russakovsky et al., 2015) is organized by the ImageNet team using a subset of the ImageNet database. The competition uses around 1.2 million images that belong to 1,000 classes (Krizhevsky, Sutskever & Hinton, 2012).

Deep networks consisting of hidden layers that learn different features at every layer are required. The deep network with the maximum amount of data can optimize learning (Lohr, 2012). However, the issue is that it is difficult to get a huge labeled dataset for training (Masud et al., 2008). Even if you get the dataset it might take a substantial amount of time and cost to train a deep network. This is where the concept of pre-training (Erhan et al., 2010; Zoph et al., 2020) comes in. Notably, many models that have been trained on powerful GPUs using millions of images for hundreds of hours are available. Existing pretrained models have been processed on a sizeable dataset to solve problems similar to ours (Marcelino, 2018). We performed transfer learning after selecting the pretrained model to use. Transfer learning transfers information from one domain to another and improves the learner (Pan, Kwok & Yang, 2008; Torrey & Shavlik, 2010; Weiss, Khoshgoftaar & Wang, 2016). We have two domains, namely, source DS and target DT with their tasks TS and TT respectively. Transfer learning is defined as the process of improving the target predictive function given domains and tasks by using the required information from DS and TS, where DS is not equal to DT or TS is not equal to TT (Weiss, Khoshgoftaar & Wang, 2016). We used five architectures available in the Keras (Ketkar, 2017) library, namely, VGG16, VGG19, ResNet50, InceptionV3, and Xception (Sarkar, Bali & Ghosh, 2018). We also aim to compare the performance of each architecture on our chosen dataset. Many applications of scene recognition are used in social media and the tourism industry to extract the location information of images. Hence, objectives of this study are as follows:to investigate the different architectures that will be used with ImageNet weights to perform the task of multiclass classification;

to compare the performance of different architectures used to perform the task of transfer learning on the dataset consisting of multiple classes;

to solve the scene recognition task using ImageNet and then use it for multilabel scene classification;

to explore the performance of transfer learning in the problem using metrics and graphs obtained;

to design an ensemble using optimally performing models and improve image classification.

The remainder of this paper is arranged as follows. The literature is reviewed in “Literature Review”. The methodology and dataset used in this study are introduced in “ENSEMV3X: Ensemble model of InceptionV3 and Xception”. Learning curves are described in “Results”. The results of this study are discussed in “Discussion”. Finally, the conclusion and future scope are presented in “Conclusions and Future Scope”.

Literature Review

LeNet-5 (LeCun et al., 1989) mainly describes convolutional neural networks (CNNs) as a stack of convolutional layers, followed by fully connected layers that have been successfully used on various datasets, such as MNIST, (Deng, 2012) and in the ImageNet classification challenge. ImageNet is a large and diverse dataset widely used in machine vision tasks. ImageNet Large Scale Visual Recognition Challenge (ILSVRC) is a challenge that is organized by the ImageNet team that uses a subset of the ImageNet database i.e. 1.2 million images belonging to 1,000 object classes. Krizhevsky, Sutskever & Hinton (2012) trained a deep CNN ILSVRC-2010 (Berg, Deng & Fei-Fei, 2010) to categorize images belonging to the dataset and achieved top-1 and -5 error rates of 37.5% and 17.0%. The methodology used a CNN comprising of 650,000 neurons and around 60 million parameters.The deep network used the AlexNet architecture, which consists of five convolutional layers, followed by three fully connected layers with a final 1,000-way softmax layer. The researchers also used two GPUs to train the network given that 1.2 million images are excessively large for a single GPU.

Simonyan & Zisserman (2014) addressed depth, with is another very important facet of CNNs. The researchers evaluated deep networks consisting of 19 layers known as VGG19, which is composed of a stack of convolutional layers, followed by three fully connected layers and a softmax layer as the final layer. VGG19 achieved 23.7% and 6.8% as the top-1 and -5 validation error rates, respectively, and 6.8% as the top-5 test error rate.

Yosinski et al. (2014) in the year 2014 used a neural network trained on ImageNet and attempted to show the transferability of features and subsequently demonstrated that feature transferability decreases as the distance between initial and final tasks increases.

Le & Yang (2015) and Yao & Miller (2015) attempted to investigate the effect of various parameters, such as convolutional layer depth, dropout layers, and receptive field size, on the accuracy in the Tiny ImageNet Visual Recognition Challenge. The TinyImageNet dataset is a subset of the dataset used in ILSVRC-2010 that consists of 200 object classes, which use 500 training images and 50 validation and testing images. The study also increased the network depth and then applied techniques, such as parametric rectified linear unit (PRELu) (Xu et al., 2015) and dropout (Srivastava et al., 2014), to the model. The researchers used images that were not initially annotated whereby algorithms must suggest labels to which images belong. The methodology achieved a final error rate of 0.444.

Szegedy et al. (2015) proposed a new architecture called Inception for ILSVRC 2014. The Inception architecture mainly consists of modules that are stacked one over the other and sometimes succeeded by max-pooling layers. The researcher also suggested the use of GoogLeNet, which is another aspect of the Inception architecture that consists of 22 layers and demonstrates a top-5 error rate of 6.67%.

He et al. (2015) proposed PReLU, which is a generalization of the standard rectified linear unit (ReLU). PReLU improved the model fitting without any extra cost and helped achieve a top-5 test error rate of 4.94%, thereby indicating a 26% improvement over GoogLeNet (winner of ILSVRC14).

Han et al. (2015) followed three steps and attempted to only maintain important connections and weights without degrading the accuracy. The researchers used the ImageNet dataset with AlexNet and VGG-16 Caffe (Jia et al., 2014) models to show how reduced the number of parameters 9 and 13 times without deteriorating the accuracy.

Sun (2016) implemented a different version of ResNet with 34 layers. Huang et al. (2016) developed an improved model as the baseline after applying data augmentation and stochastic depth and then compared the results with those of other residual networks. Bloice, Stocker & Holzinger (2017) showed that heavy image augmentation significantly improves the accuracy with an error rate of 34.68%.

Simon, Rodner & Denzler (2016) put forward a set of pretrained models, including ResNet-10, ResNet-50, and batch normalized versions of AlexNet and VGG19. Such models can be trained within minutes using powerful GPUs (Akiba, Suzuki & Fukuda, 2017). All their models performed better than previous models with Top-1 and -5 error rates, which are equivalent to 26.9% and 8.8% for VGG19, respectively.

Huh, Agrawal & Efros (2016) investigated the importance of ImageNet in learning acceptable features and explored the behavior of ImageNet by fine-tuning three tasks, namely, object detection using PASCAL VOC (Everingham et al., 2010) 2007 dataset, action classification on PASCAL-VOC 2012 dataset, and scene classification on the SUN dataset (Xiao et al., 2010).

Bastidas (2017) benchmarked on the Tiny ImageNet challenge by adapting and fine tuning two such models, namely, InceptionV3 and VGGNet (Wang et al., 2015).

Transfer learning extracts features from trained ImageNet networks. Extracted features are further used to train SVMs (Mayoraz & Alpaydin, 1999) and logistic regression classifiers (Donahue et al., 2014; Sharif Razavian et al., 2014; Smola & Schölkopf, 1998) and showed great performance on tasks that were different from ImageNet classification. Kornblith, Shlens & Le (2019) showed that ImageNet architectures generalize appropriately across datasets by using 16 networks with a top-1 accuracy range of 71.6% to 80.8% in ILSVRC 2012.

Our study aims to address the following research gaps and issues:Transfer learning improved the scene classification accuracy (Akilan et al., 2017) but some problems were still identified during analysis.

– A considerable amount of training data are required when the deep CNN model containing a large number of parameters is trained using transfer learning.

– Satisfactory results are difficult to obtain due to the high complexity of rich and dense indoor images.

Images with rich semantic content pose a problem (Liu & Tian, 2019) when attempting to solve the scene classification problem.

Ensemv3x: ensemble model of inceptionv3 and xception

Face recognition (Jain, Nayyar & Bachhety, 2020), object detection (Goyal & Benjamin, 2014), and scene classification are application areas of CNN (Khan et al., 2018; Koushik, 2016; O’Shea & Nash, 2015). CNN comprises three layers, namely, convolutional layer, pooling layer and a fully connected layer. CNNs use the concept of learnable kernels in the convolutional layer to form the base for CNN. The convolutional layer produces a 2D activation map (Krizhevsky, Sutskever & Hinton, 2012) and as per the stride value, we glide through the input to produce a scalar product for each value in the kernel. Forward pass of convolutional layer (Liu et al., 2015) can be described by Eq. (1).

(1) xi,jl=∑m∑nwm,nloi+m,j+nl−1+bi,jl,

where x is the neuron output for row i and column j, w represents the weights, and b represents the biases (Gordon & Desjardins, 1995) for the lth convolutional layer. The number of parameters is reduced with the help of pooling layers. The subsequent application of an activation function (Sibi, Jones & Siddarth, 2013) accelerates up the learning process by adding non-linearity. Maxout, tanh, ReLU, and variants of ReLU, such as ELU, leaky ReLU, and PReLU (Xu et al., 2015), help add nonlinearity. Fully connected layers (Basha et al., 2020) are then used to help determine optimal weights using backpropagation. These layers take the input from previous layers and analyze the output of all previous layers (Khan et al., 2019). Fully connected layers perform linear and nonlinear transformation on the incoming input. Equation (2) denotes the equation for linear transformation.

(2) z=WT⋅X+B,

where X represents the input; W is the weight; and B is the bias, which is a constant.

We investigate and implement different ImageNet architechtures, such as VGG16, which was used to win the ILSVRC 2014, in this study. It uses a large number of hyperparameters of approximately 138 million and a 3 × 3 filter for a stride of one for convolutional layers and a 2 × 2 filter for a stride of two for the maxpool layer. This structure is followed throughout the architecture. The number 16 in this study denotes the number of layers with weights. Figure 1 shows the architecture of VGG16 consisting of two convolutional blocks with two layers each, three convolutional blocks with three layers, and two fully connected layers. The number of filters increases with the increase of convolutional blocks. Hence, the first block has 64 filters with a size of 3 × 3, the second block has 128 filters, the third block has 256 filters, and the last two blocks have 512 filters with a size of 3 × 3. The VGG16 architecture used in this study classifies the output into 16 classes.

Figure 1 VGG16 architecture.

Simonyan & Zisserman (2014) further suggested that VGG19 is a deep network with a large number of layers. The image input with a size of 224 × 224 for this model was trained on more than a million images belonging to 1,000 object categories. VGG19 successfully learned many features from a wide range of images. Figure 2 depicts the VGG19 architecture consisting of five blocks of convolutional layers, followed by two fully connected layers. The number of filters for the first block are 64 followed by 128 and 256 filters, and the last two blocks have 512 filters with a size of 3 × 3. Figure 2 is the general representation of the VGG19 architecture.

Figure 2 VGG19 architecture.

ResNet50 is generally used as the model for transfer learning and a miniaturized version of ResNet152. ResNet (He et al., 2015) was the winner of the ImageNet challenge in 2015 and has since been used for a number of tasks related to computer vision. This model helps to overcome the problem of vanishing gradient (Hochreiter, 1998) and allows us to train neural networks that are more than 150 layers deep. Skip connection (Furusho & Ikeda, 2019) is the main innovation in ResNet that allows the skipping of some irrelevant layers for training.

Szegedy et al. (2015) introduced the Inception model for ILSVRC14. Inception V3 is a CNN with 48 layers given by Google and trained on nearly 1 million images that classifies images to 1,000 classes, such as mouse, pencil, keyboard and a number of animals. Figure 3 depicts the architecture of the Inception model. Convolutional layers (1 × 1, 3 × 3, and 5 × 5) are used and their output is concatenated to form the input in the next stage. Convolutional layers with a size of 1 × 1 are used in this study to reduce the dimensionality (Van Der Maaten, Postma & Van den Herik, 2009). This inception module is thus used in large architectures. Such a concept is mainly used to highlight the important learnings from previous layers for all subsequent layers (Stone & Veloso, 2000).

Figure 3 Inception architecture.

Finally, Xception is another CNN that is 71 layers deep and requires an image with an input size of 299 × 299. Chollet (2017) proposed an original deep CNN inspired by Inception and named it Xception. Xception performs better than InceptionV3 on the ImageNet dataset and performs significantly better on large image classification datasets with nearly 350 million images and 17,000 classes. Figure 4 depicts the architecture of the Xception model.

Figure 4 Xception Architecture.

Dataset

We used a dataset published by Intel to host an image classification challenge. The dataset extracted from Kaggle and is called “Intel Image Classification,” (Intel, 2018) consisting of 25,000 images under six labels, namely, building, glacier, mountain, forest, sea, and street. We utilized the entire dataset for training the model and subsequently tested it. The dataset has around 14,000 images for training, 3,000 for testing, and 7,000 for prediction.

Methodology

We used the concept of transfer learning in this study. Transfer learning (Weiss, Khoshgoftaar & Wang, 2016) helps build deep neural networks in a fast and accurate manner due to the usage of pretrained networks that can later be used for specific tasks.

CNNs consist of a convolutional base and a classifier. The convolutional base helps extract features from images using pooling and convolutional layers while the classifier helps label the image on the basis of recognized features. Figure 5 shows the two components of CNNs. Keras comes bundled with a number of models, such as VGG, ResNet, Inception, Xception (Sarkar, Bali & Ghosh, 2018), and others. These models generally have two parts, namely, model architecture and weights (Gulli & Pal, 2017). Model weights are excluded from Keras but can be included using the weight parameter in the model definition. We used the weight parameter in the model definition to import ImageNet weights for each model during implementation. Figure 6 shows the flowchart for the implementation. We performed the task of multiclass classification (Aly, 2005) to analyze deep features of various scenes, such as forest, sea, or street, using models bundled with Keras. We started with a pretrained model by importing it from the Keras library and then setting the weight parameter in the model to “ imagenet.” Images were converted to appropriate dimensions according to the model and then features were extracted from images to form a part of the convolutional base. We used fully connected layers as the classifier to classify images to their respective classes depending on the probability of each class.

Figure 5 Components of CNN.

Figure 6 Proposed methodology.

The top two models, Xception and InceptionV3, were chosen for EnsemV3X after obtaining the results. The two models are used to extract image features from the dataset, and their individual weight files are saved after training images. These weight files are then loaded into the respective classifiers of the two architectures and, the models are used to create an ensemble. Figure 7 shows the ensemble model created from Xception and InceptionV3. Weight files obtained by running the models for InceptionV3 and Xception were then used to train the ensemble and obtain the results for the test data using the ensemble model.

Figure 7 EnsemV3X.

Results

Learning curves (Amari, 1993) obtained after running the models are important sources for measuring the performance of the deep learning model. Training and validation curves are two types of learning curves calculated from training and validation datasets (Xu & Goodacre, 2018) respectively. Loss (Hastie, Tibshirani & Friedman, 2009) helps us understand how bad a model performs with every epoch and the extent of its performance in a particular data sample. Meanwhile, accuracy helps denote how accurate the model prediction is to the actual data. Training and the validation curves for each of the architecture are demonstrated. On the one hand, the plot for training accuracy and training loss shows the outcome for the training data. On the other hand, the plot for validation accuracy and loss show the outcome for the validation data. Validation data are used to validate outcomes from the training data and check what the model has learnt from the data. The distance between curves shows the performance of the model.

Figure 8 illustrates the training versus validation accuracy and loss curve for the InceptionV3 model. Similarly, Fig. 9 presents the same curves for the ResNet50 model. VGG16 and VGG19 training versus validation accuracy and loss curves are depicted in Figs. 10 and 11, respectively. Figure 12 shows the training versus validation accuracy and loss curve for the Xception model.

Figure 8 Training and validation metrics for InceptionV3.

(A) Training vs validation accuracy and (B) Training vs validation loss.

Figure 9 Training and validation metrics for ResNet50.

(A) Training vs validation accuracy and (B) Training vs validation loss.

Figure 10 Training and validation metrics for VGG16.

(A) Training vs validation accuracy and (B) Training vs validation loss.

Figure 11 Training and validation metrics for VGG19.

(A) Training vs validation accuracy and (B) Training vs validation loss.

Figure 12 Training and validation metrics for Xception.

(A) Training vs validation accuracy and (B) Training vs validation loss.

Discussion

Accuracy (Hossin & Sulaiman, 2015) is defined as the number of correct predictions made by our model to the total number of predictions. Eq. (3) represents the formula for accuracy.

(3) A=1n∑i=1n|Mi∩Ni||Mi∪Ni|,

where Mi represents the given labels in the dataset, Ni represents the predicted labels in the dataset, and A denotes accuracy.

These terms can be understood in the context of a confusion matrix (Chow, Dom & Lin, 2013). The confusion matrix is a metric that uses four combinations of predicted and actual values. True positive represents true terms that were predicted to be true. True negative is a false term that was predicted to be false. False positive is a term with a negative actual class that was predicted to be positive. False negative is a term with a positive actual class that was predicted to be negative.

VGG16 and VGG19 confusion matrices are shown by Figs. 13 and 14, respectively. Figures 15, 16, 17, and 18 depict the confusion matrices for the Xception model, ResNet50 model, InceptionV3 model, and Ensemble models, respectively. These confusion matrices represent the outcome of around 3,000 images used for testing. Similarly, other metrics can also be calculated according to the confusion matrix obtained for each model. The label 0 denotes one of the six classes and the same applies for other labels. The confusion matrix shows how many images were labeled class 0 and actually belonged to class 0 and those that were incorrectly classified. Precision (Hossin & Sulaiman, 2015) and recall (Hossin & Sulaiman, 2015) are expressed as follows:

Figure 13 Confusion matrix for VGG16.

Figure 14 Confusion matrix for VGG19.

Figure 15 Confusion matrix for Xception.

Figure 16 Confusion matrix for ResNet50.

Figure 17 Confusion matrix for InceptionV3.

Figure 18 Confusion matrix for EnsemV3X.

(4) P=1n∑i=1n|Mi∩Ni||Ni|,

(5) R=1n∑i=1n|Mi∩Ni||Mi|,

where Mi represents the labels given in the dataset and Ni represents the labels predicted for the instance i. Equation 4 depicts the formula for precision and Eq. (5) depicts the formula for recall. Lastly, Eq. (6) depicts the equation for f1-score (Hossin & Sulaiman, 2015) can be calculated using precision and recall as follows:

(6) F1=1n∑i=1n2|Mi∩Ni||Mi|+|Ni|.

The comparison of performance of all models is presented in Table 1.

Table 1 Performance metrics.

Model	Accuracy (%)	Precision (%)	Recall (%)	F1-score	
InceptionV3	89	89	89	89	
ResNet50	37	32	37	29	
VGG16	89	89	89	89	
VGG19	87	87	87	87	
Xception	90	90	90	90	
EnsemV3X	91	91	91	91	

Conclusions axnd future scope

ImageNet, a dense and diverse dataset that has been trained with over 22,000 object categories on the basis of WordNet, has demonstrated excellent performance in object recognition but poor performance in scene recognition. We attempted to perform the task of multilabel scene recognition using ImageNet architectures from the Keras library and Google Colab and then compared their performance, as listed in Table 1. The results showed that ImageNet can achieve satisfactory results and further improve by changing architectures depending on the problem. ImageNet can be used to solve many scene recognition challenges given its very diverse dataset. We also attempted to prepare an ensemble using models previously trained with ImageNet. The InceptionV3 and Xception models obtained the best performance compared with the other models and demonstrated an accuracy of 91% after combining them to form an ensemble.

ImageNet architectures for rich and dense images captured by mobile phone cameras can be the focus of future investigations. Furthermore, ImageNet and its counterpart PlacesCNN can be compared on the same dataset to analyze their performance for both object detection and scene recognition.

Supplemental Information

Supplemental Information 1 Code for ensemble used in Multi Label Scene Classification.

The code used to run the 5 models to get the results and the code used to take Inception and Xception model and create the Ensemble-EnsemV3X.

Click here for additional data file.

Supplemental Information 2 Data used in multi label scene classification.

Click here for additional data file.

Additional Information and Declarations

Competing Interests

Author Contributions

Data Availability

The authors declare that they have no competing interests.

Priyal Sobti conceived and designed the experiments, performed the computation work, prepared figures and/or tables, and approved the final draft.

Anand Nayyar performed the experiments, analyzed the data, prepared figures and/or tables, authored or reviewed drafts of the paper, and approved the final draft.

Niharika conceived and designed the experiments, performed the computation work, prepared figures and/or tables, authored or reviewed drafts of the paper, and approved the final draft.

Preeti Nagrath performed the experiments, analyzed the data, performed the computation work, prepared figures and/or tables, and approved the final draft.

The following information was supplied regarding data availability:

The raw data and code is available in the Supplementary Files.

Additional training, testing and validation is available at Kaggle: https://www.kaggle.com/puneet6060/intel-image-classification.

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
