# Peer review of "EnsemV3X: a novel ensembled deep learning architecture for multi-label scene classification"

_PeerJ Computer Science, doi:10.7717/peerj-cs.557_

## Round 0.1 · original submission · Major Revisions

The recommendations from the two reviewers are divergent, but they both raise concerns. As a result, an essential revision and improvement is recommended.

Reviewer 1 ·

Basic reporting

This paper is about deep learning classification.
1- It is well written and well structured.
2- The literature review and the references are sufficient and updated.
3- The tables and the figures are understandable.

Experimental design

1- The novelty of this paper is not clear. The proposed structure is based on ImageNet and VGG. The authors should explain more about their collaborations.
2- The authors should explain more on the proposed structure. Based on Figure 7, the only difference between the proposed structure and ImageNet is a fully-connected NN. The authors should explain more about how Figure 8 is related to Figure 7.

Validity of the findings

1- The result and the discussion sections should be more clarified based on Figures 14-19.

Reviewer 2 ·

Basic reporting

The authors are advised to revise the manuscript thoroughly and carefully to avoid any possible technical and grammatical errors.

Experimental design

The experimental part of the article lacks the experimental design, which is a serious drawback of the experiment. Besides, the results cover a very narrow area. Meanwhile,no statistical analysis of the results.

Validity of the findings

no comment

Additional comments

Reviewer comments:
This paper presented a novel Ensembled Deep Learning Architecture for multi-label scene
classification. However, the logical relationship is confused and the core content is not prominent of the manuscript. The other issues of the manuscript are as follows:
1.The content of the abstract is not complete, and the purpose of scientific work is not highlighted.
2.In LITERATURE REVIEW, "Lack of indoor scene datasets makes it difficult to get satisfying results by fine-tuning a pre-trained CNN". Why it is difficult to get satisfying results? The satisfying results are not explained clearly.
3.The authors are suggested to provide a comparative study between the proposed work and earlier published works. It will help for future researchers.
4.What are the major advantages of proposed new model over the other available techniques, such as WideResNets, AlexNet, VGG, Inception, ResNets and others?
5.The experimental part of the article lacks the experimental design, which is a serious drawback of the experiment. Besides, the results cover a very narrow area. Meanwhile,no statistical analysis of the results.
6.The authors are suggested to improve the conclusions.
7.The authors are advised to revise the manuscript thoroughly and carefully to avoid any possible technical and grammatical errors.
8.The format of references is not uniform.
My overall opinion then, I don't believe the paper can be recommended for publication.

---

## Round 0.2 · Minor Revisions

One reviewer is happy with the manuscript, and the other raises some concerns. As a result, the paper needs a further revision.

Reviewer 2 ·

Basic reporting

Interesting article and you can see many changes. Besides, the article is generally correctly written with well-chosen bibliography.

Experimental design

In this paper, it aims to show multi-label scene classification using five architectures namely VGG16, VGG19, ResNet50, InceptionV3, and Xception using imagenet weights available in the Keras library. The experiments have shown a performance comparison of the different architectures further in the paper.

Validity of the findings

The proposed a new model, EnsemV3X with reduced number of parameters giving an accuracy of 91% better than the best performing models Inception and Xception.

Additional comments

Interesting article and you can see many changes. All the comments have been addressed clearly, so I have no other comments.

Reviewer 3 ·

Basic reporting

no comment

Experimental design

no comment

Validity of the findings

no comment

Additional comments

Authors proposed EnsemV3X by a way of dividing the feature extraction with weight files and feature recognition with using five classical architectures. It is an interesting idea. There are some concerns:
1. Different convolution kernels will lead to different effects of feature extraction. How to choose the appropriate weights as a general feature because the weights files are best for the default models with special kernels, but there is no guarantee that they will be universal?
2. Authors make an ensemble feature recognition with fully-connection ways based on the outputs of five models. It still needs the weights of connection. How to determine these weights in order to get the best results?
3. The definition of loss should be shown in order to be clear.
4. Fig.9 to 11 are not clear. For example it seems the training accuracy has not be stable at 70 epochs. What is the end condition? Do the accuracy and the loss have the same coordinates? What is the relation of them? More explanations are necessary. The similar problems are for figure 13 to 18.

---

## Round 0.3 · accepted · Accept

The paper has been revised and is ready to be accepted.